# Effectiveness and feasibility of structured emotionally focused family therapy for parents and adolescents: Protocol of a within-subjects pilot study

**Henk Jan Conradi** [1]*, **Daphne Meuwese**[2], **Lenny Rodenburg**[3], **Pieter Dingemanse**[2], **Trudy Mooren**[4,5]

1 Department of Clinical Psychology, University of Amsterdam, Amsterdam, The Netherlands, 2 Altrecht Institute for Mental Health Care, Utrecht, The Netherlands, 3 Psychologiepraktijk Lenny Rodenburg, Bussum, The Netherlands, 4 Foundation Centrum '45, Arq Psychotrauma Expert Group, Diemen, The Netherlands, 5 Department of Clinical Psychology, Utrecht University, Utrecht, The Netherlands

* h.j.conradi@uva.nl

**Data Availability Statement:** No datasets were generated or analysed during the current study. All

## Abstract

Mental health issues are widespread among children and adolescents worldwide. Although mental health difficulties may manifest themselves in many different diagnoses, there is growing support for a limited number of underlying transdiagnostic processes. Attachment encompasses a key transdiagnostic mechanism, namely emotional regulation. This study protocol aims to evaluate the feasibility and potential effectiveness of structured emotionally focused family therapy (EFFT), the goal of which is to develop secure attachment between parents and their children to reduce children's vulnerability to mental health problems. A within-subjects design with three waves, a waiting period, treatment, and follow-up, will be conducted. Families will serve as their own controls. Approximately 15 to 20 families with adolescents (aged 12−18 years) as the 'identified patients' will be included. They will participate in 16−21 sessions of EFFT. The study will use a multi-method approach. Self-report questionnaires will be administered repeatedly (i.e., pre-waiting period, pre-treatment, half-way treatment, post-treatment, and follow-up), measuring parent-adolescent attachment, partner-partner attachment, negative interactions, and adolescent psychological complaints. Multi-level analyses will be conducted. Semi-structured interviews will be administered at follow-up to evaluate feasibility and acceptability of EFFT. Treatment integrity will be assessed. The present study is the first to evaluate feasibility of structured EFFT and obtain a first impression of its effectiveness. This information will help us to improve EFFT. Limitations are discussed.

**Trial registration:** Recruitment commenced in June 2022. The approximate trial duration is 36 months. The trial was registered at ClinicalTrials.gov (NCT05657067) on December 9, 2022, and Open Science Framework (https://osf.io/39dt2/) on June 14, 2022.

relevant data from this study will be made available upon study completion.

**Funding:** The authors received no specific funding for this work.

**Competing interests:** PD is a member of the board of EFT Netherlands. LR provides EFFT. This does not alter our adherence to PLOS ONE policies on sharing data and materials. The other authors declare that they have no competing interests.

**Abbreviations:** ARE, Accessibility, Responsiveness, Emotional Engagement questionnaire; CSI-4, Couple Satisfaction Index; ECR-RS, Experiences in Close Relationships—Relationship Structures questionnaire; EFFT, Emotionally Focused Family Therapy; RDS, Relationship Dynamics Scale; SEPTI-TS, Self-Efficacy for Parenting Tasks Index Toddler Scale; SDQ, Strengths and Difficulties Questionnaire.

# Introduction

The prevalence of mental health problems of children and adolescents worldwide is substantial at 10–20% [1]. These mental health difficulties may manifest themselves in many different diagnostic categories, such as depression, anxiety, withdrawal, addictions, and conduct-related problems. However, there is growing support that underlying this multiplicity is a limited number of transdiagnostic processes that cut across diagnostic boundaries set by psychiatric nosologies like the *Diagnostic and Statistical Manual of Mental Disorders* (DSM) [2]. Attachment is considered such a process [3] encompassing a key transdiagnostic mechanism, namely emotional regulation. Secure attachment relations of children with their attachment figures (i.e., parents or caregivers), serve as a source of functional emotional co-regulation, whereas insecure attachment relations are related to dysfunctional emotional regulation [4]. Emotionally focused family therapy (EFFT) aims at the development of secure attachment between parents and their children to reduce children's vulnerability to mental health problems [5]. In this way, EFFT facilitates the development of fundamental resilience for children characterized by functional emotional regulation, enhanced self-esteem, and trust in others through which they can thrive [5].

Secure attachment is characterized by healthy emotional regulation. Securely attached people expect attachment figures to be available and responsive, and functionally activate their attachment system by seeking proximity and openly asking attachment figures for help in fulfilling their needs of validation (being loved and appreciated when feeling insecure), support (receiving help when experiencing emotional problems), and reassurance (being comforted when distressed) [4]. The help of available and responsive attachment figures results in functional co-regulation of emotions and serves as a resilience factor in the maintenance of mental health [6].

By contrast, insecure attachment develops when attachment figures are not, or are inconsistently, available, and responsive. Persons who are unsure whether attachment figures are available and responsive when they need them are inclined to *hyperactivate* their attachment system to overcome fear of being rejected. Hyperactivation is characterized by an amplification of emotions to draw attention from their attachment figures, clinging to them and tending to coerce them to offer validation, support, and comfort. When they do not get what they want, anger may surface [4]. Conversely, people who are sure their attachment figure will be unavailable and unresponsive tend to *deactivate* their attachment system to avoid being rejected. Deactivation is characterized by avoidance of others, not asking for support but instead relying on oneself as source of emotional regulation, mainly by suppressing emotions [4]. Both hyper- and deactivation of the attachment system result in dysfunctional emotional regulation and are risk factors for the development of psychopathology, for example, anxiety [7, 8], depression and suicidal behavior [9], substance abuse, and criminal behavior [10].

Although attachment is relatively stable throughout the life span [11] it is also amendable to change by therapy [12, 13]. Three family therapies with attachment as main treatment goal are available, namely emotionally focused family therapy (EFFT) [5], attachment-based family therapy (ABFT) [14], and dyadic developmental psychotherapy (DDP) [15]. Although all three therapies share the development of secure attachment between parents and children as the main treatment goal, there are also important differences reflecting their particular strengths. EFFT has a strong emphasis on the *experiencing* of emotions throughout the therapy sessions. This is achieved by enactments, the so-called 'EFT tango' [16], in which emotions are basically experienced as bodily sensations and are explored for their meaning and the associated unmet attachment needs. This emotional experiencing is seen as the most important agent of durable change. Verbalization of emotions (expressed cognitions) are important in EFFT to

communicate emotions to others but are not seen as the principal mechanism of change [17]. A noticeable strength of ABFT is the clear *structure* it offers by prescribing sessions with the family as a whole and specific subsystems. Working with subsystems reduces the complexity inherent to family interventions and thus increases transparency and the chance of treatment success. Finally, DDP stresses the importance of the *attitude* of the therapist, named 'PACE' (playful presence, acceptance, curiosity, and empathy), which is particularly important when working with children. Based on EFFT as described by Furrow et al. [5], we incorporated the strengths of ABFT (more structure) and DDP (attitude) in our adaptation of EFFT. Of note, *Emotionally*-focused family therapy, the intervention examined in this study, should not be confused with *Emotion*-focused family therapy [18] Emotionally- focused family therapy has attachment as its central focus, whereas Emotion-focused family therapy works with the general emotion coaching principles as described by Greenberg.

The rationale behind EFFT [5] is that mental health problems in children and adolescents often originate and/or are exacerbated by negative interaction patterns rooted in insecure attachment bonds within families. Therefore, the main treatment goal of EFFT is the development of secure attachment between parents and their children. On the one hand, this means that parents are trained to be available and responsive to their children's attachment needs of validation, emotional support, and comfort; in other words, they are taught to act as a secure base. On the other hand, the children are stimulated to seek proximity to their parents and openly express their attachment needs based on increased trust in the accessibility and responsiveness of their parents. Enhancing secure attachment will reduce negative interactions, buffer related stress and other psychological complaints, and will strengthen resilience. This makes EFFT a promising systemic transdiagnostic intervention. EFFT can be applied with parents and their children aged six years and older, for children younger than six it would be difficult to express their attachment needs verbally.

The current pilot study is designed as a first test of the newly developed protocol of structured EFFT. Therefore, it is important to evaluate whether such treatment is desirable and feasible, and establish an initial indication of its effectiveness. Evaluation of the feasibility of EFFT will be based on semi-structured interviews with family members who participated in order to obtain their evaluation of (specific phases of) EFFT and to identify possible points of improvement of EFFT. Concerning the effectiveness of EFFT, only one early pilot study is available that has shown EFFT to be a promising intervention for families with children suffering from bulimia [19]. However, we expect EFFT to be effective since emotionally focused couples therapy (EFCT), of which EFFT is a fundamental extension, is an empirically supported treatment [20]. In this study we will examine the effectiveness of structured EFFT with regard to several outcomes. We chose escalation in negative interactions between child, mother and father as the primary outcome measure as the transdiagnostic idea that the child's problems are originating in and/or exacerbated by negative interactions within the family is central to EFFT. Other equally important outcome measures are accessibility and responsiveness of the parents to the child's and their partner's needs, attachment between the parents and the child, and child psychological complaints. Additional outcome measures are: disciplining of the child by the parents; attachment between the parents as partners, and parental relationship satisfaction. Concerning all outcomes, we hypothesize:

1. less change during the waiting period compared with the treatment phase;

2. gain during the treatment phase; and

3. substantial maintenance during follow-up.

## Method

### Design

A within-subjects design with three waves will be used. The waiting period will span approximately two months, the treatment phase will last three to four months, followed by a two months follow-up period concluded with a booster session. A randomized controlled design was deemed not feasible for this pilot study, we therefore opted for a within-subjects design in which families will serve as their own controls. By comparing change during the waiting period with change during the treatment phase, we probably get an impression of spontaneous remission vs. treatment-related change.

The study will use a multi-method approach. Evaluation of the effectiveness of EFFT will be based on self-report questionnaires, whereas evaluation of feasibility will be based on semi-structured interviews, self-reported ratings by the family members and the registered treatment attendance rates. Finally, adherence to the treatment protocol by the therapists is based on the roadmap for structured EFFT (S1 File).

### Participants

For this pilot study we will include families with adolescents aged 12–18 years (the so called 'identified' patient) who are coping with mild problems. The latter means that we will exclude (a) blended families because of the more complex loyalties that exist between children and stepparents, (b) families of which individual members cope with serious trauma such as sexual and physical abuse and severe neglect, and (c) families of which the parents or children are diagnosed with severe DSM disorders (substance abuse or psychosis) or are in therapy to maintain sufficient research sample homogeneity for this pilot study. Exclusion criteria will be determined with unstructured clinical interviews by experienced clinicians during the screening. Families will be recruited by five private EFFT practices in the Netherlands. Prior to entering the study participants are asked to give their informed consent, whereas parents provide consent for their children under the age of 16. For this pilot study we aim for 15–20 included families who will be treated by five family therapists supervised by one International Centre for Excellence in Emotionally Focused Therapy (ICEEFT) certified psychotherapist. The study was approved by the Ethics Review Board of the University of Amsterdam (2022-CP-15102) on June 10, 2022. The trial is registered at ClinicalTrials.gov (NCT05657067) on December 9, 2022, and Open Science Framework (https://osf.io/39dt2/) on June 14, 2022 and will be conducted from June 2022 onward.

### Outline of the treatment protocol and treatment adherence

The protocol-driven intervention is an adapted version of the official EFFT program as described by Furrow et al [5]. The adapted version was developed by LR, DM and HJC assisted by TM and PD, whereas Elien van Oostendorp and Marij Peeters initially played an important role. We adapted the EFFT program by applying more structure in order to make it more suitable for research purposes (see S1 File. Roadmap structured EFFT). The original EFFT program defines three global treatment phases and leaves it up to the therapist to decide to work with the family as a whole (i.e., the parents, the identified adolescent patient, and the siblings), or with subsystems (i.e., the parents as partners, or the parents and the identified adolescent patient), or with individuals (i.e., one of the parents or the identified adolescent patient, separately). To reduce the complexity of systemic family therapies like EFFT we opted for a structured protocol in which we prescribe four distinct treatment phases and the specific (sub-) systems that are targeted. This structured EFFT protocol has two advantages: (1) it enables us

to put the effectiveness of a well circumscribed intervention protocol to the test, and (2) it is easier to transfer to novice therapists who are less experienced in handling the complexity of family systems.

Structured EFFT consists of four treatment phases aiming at nine treatment objectives in 16–21 protocolized sessions. Phase 1 consists of one or two sessions with the whole family in which the development of a working alliance is started, interaction problems between family members are observed and assessed, and awareness of escalation of interaction patterns is strengthened. Finally, the adolescent's problems are reframed as originating in, and/or exacerbated by *relational problems* between the parents and the child. Consensus between the family and the therapist about recovery from negative interaction patterns as the treatment goal is crucial.

In phase 2 the parents (phase 2A) and the adolescent (phase 2B) are separately seen by the therapist. Phases 2A and B are conducted in parallel fashion. The overarching goal in phase 2 is deepening of emotional experiences during negative interactions through enactments. Deepening is achieved by exploring the non-expressed *core emotions* like fear about rejection or sadness about the lost connection that *underlie the visible reactive behavior* like anger or emotional avoidance related to withdrawal behavior. The core emotions refer to the unfulfilled attachment needs of validation, support, and reassurance.

Phase 2A (four to six sessions) with the parents only is complex as parental insecure attachment strategies are explored at three relationship levels. First, the attachment relationships between each of the parents and their own parents are examined to understand the origins of their own insecure (reactive) attachment behavior. In this way self-blame for the insecure attachment relation with their own adolescent can be reduced. Moreover, emotional re-experiencing of their own insecure attachment history will facilitate the parent's understanding of their adolescent's insecure attachment behavior, pain, and unfulfilled needs. Second, the relationship of the parents as partners is explored and the attachment bond between them made more secure to create: (a) a secure *role model* of functional emotional co-regulation for their children, (b) a secure attachment base for their children, and (c) a stronger parental alliance that is more united and transparent in the rearing (including discipline) of the adolescent. Third, the relationship between the parents and the child, the main focus of EFFT, is explored. The purpose is preparing the parents to be more available and responsive (the building blocks for secure attachment relations) to the attachment needs of their adolescent.

In phase 2B (three sessions) with the adolescent alone, an imagined enactment of a painful interaction with the parents will be set up to evoke an experience of non-expressed core emotions like fear about rejection, sadness about the lost connection, or bottled-up anger, that underlie the visible reactive behavior like expressed anger or emotional numbing related to avoidance behavior. The core emotions will be used as a portal to unfulfilled attachment needs of validation, support, and reassurance. Finally, the adolescent will practice the secure attachment strategy of openly expressing the experienced attachment needs instead of hyperactivation or deactivation.

In phase 3 (three sessions) the preparation work done in phase 2A with the parents (practicing being available and responsive to their child's attachment needs) and phase 2B with the child (practicing openly expressing those needs) come together; therefore, the subsystem of the parents and the adolescent rejoin. The therapist sets up an enactment in which parents and adolescent work on the development of a secure attachment bond. This is done by the adolescent's sharing of painful experiences in which his or her parents were unavailable and unresponsive to their needs. The parents react by acknowledging their mistakes and offer their apologies. Subsequently, the adolescent expresses his or her need for parental validation, support, and/or consolation while the parents are accessible and responsive. To reach durable

change, it is important that this sharing is emotionally felt, and the enactment is repeated several times.

Finally, in phase 4 (three sessions) the whole family attends. Consolidation of the new secure attachment interactions is central. Based on the new secure interactions, which are supposed to act as a source of resilience and a buffer for stress, individual psychological problems like autism, attention deficit, substance abuse, and/or more practical problems like school-related issues, are addressed. When needed, additional interventions will be agreed upon. Treatment is finished with a booster session two months later with the whole family.

Adherence to the treatment protocol by the therapists will be examined by rating audio recordings of random sessions by independent raters by means of predefined, structured implementation checklists.

## Procedure and assessments

As we prepared the current study as a pilot study, the number of families included will be limited. Therefore, it is important to enhance statistical power by applying five repeated assessments (t1 to t5) for the quantitative data: at pre-waiting period, pre-treatment, prior to phase 3, post-treatment, and prior to the booster session. Online and paper and pencil questionnaires will be administered. The qualitative data will be collected after the booster session (t6). The spirit schedule of enrolment, intervention, and assessments is depicted in Fig 1. Siblings will not participate in the assessments.

Two measures will be taken to safeguard response to the assessments. First, to guarantee response to the pre-waiting period assessment, the therapist instructs the family during the telephone screening that treatment will start two months from the date the pre-waiting period questionnaires are completed. Second, to make a 100% response rate to the other assessments probable, respondents will gather in the therapist's room 20 minutes prior to the specific

|  | Enrolment | Parents | | | | | Close-out | Adolescent | | | | | Close-out |
| --- | --- | --- | --- | --- | --- | --- | --- | --- | --- | --- | --- | --- | --- |
| Time point |  | t1 | t2 | t3 | t4 | t5 | t6 | t1 | t2 | t3 | t4 | t5 | t6 |
| Enrolment: |  |  |  |  |  |  |  |  |  |  |  |  |  |
| Eligibility | X |  |  |  |  |  |  |  |  |  |  |  |  |
| Informed Consent | X |  |  |  |  |  |  |  |  |  |  |  |  |
| Intervention |  |  | ▬▬▬▬▬▬ | | | | |  | ▬▬▬▬▬▬ | | | | |
| Assessments |  |  |  |  |  |  |  |  |  |  |  |  |  |
| RDS |  | X | X | X | X | X |  | X | X | X | X | X |  |
| ARE |  | X | X | X | X | X |  | X | X | X | X | X |  |
| ECR-RS |  | X | X | X | X | X |  | X | X | X | X | X |  |
| SEPTI-TS |  | X | X | X | X | X |  |  |  |  |  |  |  |
| CSI-4 |  | X | X | X | X | X |  |  |  |  |  |  |  |
| SDQ |  | X | X |  | X | X |  | X | X |  | X | X |  |
| Semi-structured interview |  |  |  |  |  |  | X |  |  |  |  |  | X |

**Fig 1. Spirit schedule of enrolment, intervention, and assessment.** t1 = pre-waiting period; t2 = pre-EFFT; t3 = pre-phase 3 of EFFT; t4 = post-EFFT; t5 = pre-booster session EFFT; t6 post-booster session EFFT. ARE: Accessibility, Responsiveness, Emotional Engagement questionnaire; CSI-4; Couple Satisfaction Index; ECR-RS: Experiences in Close Relationships—Relationship Structures questionnaire; EFFT: Emotionally Focused Family Therapy; RDS: Relationship Dynamics Scale; SEPTI-TS: Self-Efficacy for Parenting Tasks Index Toddler Scale; SDQ: Strengths and Difficulties Questionnaire.

session to complete the questionnaires. The therapist will check that parents and the adolescent complete the questionnaires on their own. Completed questionnaires will be put in an envelope by the therapist which will be sealed by the adolescent in front of the others to guarantee anonymity.

## Instruments

Evaluation of effectiveness of EFFT in this pilot study focuses on the main objectives of the therapy: (1) de-escalation of negative interaction patterns; (2) enhancement of availability and responsiveness of the parents to the adolescent and between the parents as partners; (3) development of secure attachment between the adolescent and the parents and between the parents as partners; and (4) recovery from the adolescent's psychological complaints that initiated seeking help. Additional outcome measures are: (5) enhancement of relationship satisfaction between the parents as partners; and (6) disciplining of the child by the parents. We opted for brief questionnaires measuring these concepts as several questionnaires will be completed twice during one assessment by the adolescent and/or parents (see below).

**Negative interaction patterns** will be measured by the Relationship Dynamics Scale (RDS) [21], a four item self-report questionnaire. In EFFT de-escalation, or reduction of negative interactions, is a crucial prerequisite for the development of secure attachment. The RDS items tap into escalation, invalidation, negative interpretation, and withdrawal. An example item is 'Little arguments escalate into ugly fights with accusations, criticisms, name calling, or bring up past hurts.' The RDS converges with comparable constructs and can predict changes in communication skills caused by interventions [22]. Originally, the RDS was developed for couples; therefore, we reworded the items to make them suitable for adolescent-parent relationships too. In an earlier study (under review) we found for the original version a Cronbach's alpha of 0.72 for men and 0.74 for women. The RDS will be completed twice per assessment by the adolescent (separately for the relationship with the mother and the father) and by the parents (separately for the relationship with their adolescent and their partner).

**Accessibility and responsiveness** of the attachment figures, identified by Ainsworth as necessary prerequisites for the development of secure attachment, constitute a central treatment focus in EFFT. We will assess both constructs by an abbreviated version of the Accessibility, Responsiveness, Emotional Engagement questionnaire (ARE) [23]. An example item is 'If I need connection and comfort, he/she will be there for me.' Items are scored on a 5-point Likert scale ranging from 1 (disagree) to 5 (agree). As the original ARE was devised for partners in a couple relationship, we reworded the items to adapt the scale for parent-adolescent relationships. Cronbach's alpha for the six-item scale is .90 for men and women, which is obtained from a recalculation of data earlier published [24]. Of note, this concerns the original, not the reworded, version. The parents will complete the ARE for the relationship with their adolescent. The adolescent will complete the ARE two times per assessment (separately for the relationship with the mother and the father).

Discipline may play an important role in family problems as it can be challenging for parents to be emotionally accessible and responsive to the child and discipline the child simultaneously. The Self-Efficacy for Parenting Tasks Index Toddler Scale (SEPTI-TS) [25] is a self-report questionnaire assessing the parents' ability to discipline their child. The Discipline subscale consists of seven items. An example items is 'Other parents seem to have more success with setting limits for their children than I do with my child.' Cronbach's alpha is .81 [25]. Only the parents will complete this questionnaire.

**Attachment** will be assessed with the Experiences in Close Relationships—Relationship Structures questionnaire (ECR-RS) [11]. The ECR-RS consists of nine items and two subscales,

namely Anxiety about rejection and abandonment (hyperactivation of the attachment system) and Avoidance of intimacy (deactivation). Cronbach's alphas for Anxiety are 0.90 for men and 0.88 for women, and for Avoidance 0.90 for men and 0.92 respectively [11]. Originally, the ECR-RS was devised for both parent-child relationships and couple relationships. Example items are 'I usually discuss my problems and concerns with this person' and 'It helps to turn to this person in times of need.' 'This person' will be substituted by 'mother' or 'father' for the adolescent or by 'partner' for the parents as partners. Since EFFT focuses on enhancing secure attachment between the adolescent and the mother and the father, the adolescent answers the questions twice. The parents complete the questionnaire only once for their relationship with their partner, but not for their relationship with their adolescent as the child is not an attachment figure for the parents.

**Relationship satisfaction** between the parents as partners will be measured with the four items of the Couple Satisfaction Index (CSI-4) [26], a self-report questionnaire. Although relationship satisfaction is not the main treatment target of EFFT, the parental relationship constitutes the nucleus of the family upon which family relations are built. Therefore, it is important to measure satisfaction with the parental couple relationship. An example item is 'In general, how satisfied are you with your relationship?' The CSI-4 highly correlates with the Dyadic Adjustment Scale with 0.87 [26]. Cronbach's alpha is 0.94 [26].

**Adolescent's complaints** will be assessed with the Strengths and Difficulties Questionnaire (SDQ) [27]. We will administer the version for children aged 4–17 years to be completed by the adolescent, and the parent version for the parents. We will use the total score, labeled 'Total difficulties', which is made up of 20 items and has a Cronbach's of 0.81 [28]. Optional in low-risk or general population samples is splitting Total difficulties into two subscales, namely Internalizing problems (example items are 'Many fears, easily scared' and 'Rather solitary, tends to play alone') and Externalizing problems ('Restless, overactive' and 'Often has temper tantrums or hot tempers') which have a Cronbach's alpha of 0.73 and 0.78, respectively [29].

The main questions for the evaluation of the feasibility of EFFT are: (1) How do family members evaluate (specific phases of) EFFT?, and (2) What can we learn from their experiences? These questions will be explored in several ways. First, semi-structured interviews will take place after the booster session, with the adolescent being interviewed separately from his or her parents. These semi-structured interviews will last approximately 30 minutes. Open-ended questions will be posed based on a topics list. This list is the result of discussion within the research group. The topics list contains sections with questions about: (1) what motivated the family to seek help, (2) the effects of the therapy they experienced, (3) how the therapist was perceived, (4) their experiences during each therapy phase, (5) how they would rate the therapy, and (6) suggestions for improvement of the therapy. The researchers (DM and TM) who will conduct the interviews are trained in interviewing clients for qualitative research. Further, treatment attendance rates of the family members will be registered. Finally, feasibility of EFFT from the perspective of the therapists will be addressed during a panel discussion at the end of the project.

## Statistical analyses

As scores are not independent *within* respondents (repeated measurements) and *between* respondents (three dyads: mother and the adolescent, father and the adolescent, and the parents as partners) multi-level analyses are preferred. However, concerning multilevel models with three levels no agreed upon power analyses are available. Therefore, we estimated study power based on a repeated measurements ANOVA with GPower 3.1.9.7 [30]. We anticipate an effect size of $d = 0.7$ which is based on effect sizes obtained by Attachment-Based Family

Therapy, which is akin to EFFT (i.e., $d$ = .97 [31] and $d$ = 1.08 [32]), and the average effect size we obtained with a Couple relationship Education course based on EFT (i.e., $d$ = 0.6 [24]). We computed the required sample size to detect this effect size of $d$ = 0.7 with a power of .8 an alpha of .05 and a correction for sphericity of e = .5. It was calculated we would need $n$ = 20 families. This is a conservative power-estimation based on the number of families as if we only had one person in each family reporting on the outcome measures. However, as several of our outcomes are dyadic by nature (e.g., outcomes regarding interaction patterns between mother and the adolescent, father and the adolescent, and father and mother as partners) the actual power will be higher as the number of individuals will be twice as high for these outcomes.

Following the power analysis we first plan to run a repeated measurements ANOVA to test our main hypothesis concerning change during treatment plus follow-up on the primary outcome. Subsequently, Linear Mixed Models (LMM) in SPSS will be applied with repeated measurements (level 1) nested within respondents (level 2) and, depending on the specific outcome, respondents nested within the dyad of interest (level 3) [33]. In this way we will be able to assess change over time. An advantage of LMM is the possibility to use cases with partial missing data. To adjust for interdependence of the repeated measurements, the AR1 covariance structure will be applied, and to adjust for interdependence of respondents, we will include an intercept at level 3. When results of the multilevel model converge with the repeated measures ANOVA, trust in the robustness of findings will be enhanced.

We will examine change during the waiting period (comparison of pre-treatment versus pre-waiting period assessments), change during treatment (post-treatment versus pre-treatment assessments), change during follow-up (booster session versus post-treatment assessments), and change during the whole study period (follow-up versus pre-waiting period assessments). We will compute effect sizes (Cohen's $d$s) using, per comparison, the estimated marginal means, and the pooled standard deviations of the corresponding raw means [34]. Effect sizes will be interpreted as small when Cohen's $d$ is 0.35 or below, moderate when 0.36 −0.65, and large when 0.66 or higher. [35]. To evaluate missingness at random, we will apply pattern-mixture models for the dependent variables [36].

The feasibility of EFFT will be evaluated with multiple methods. First, the widely used MAXQDA software will be applied for the qualitative analyses of the semi-structured interviews. We will follow the steps described by Boeije [37]. for inductive thematic analysis. This starts with open coding: fragments relevant to answering the main question will be labeled. Then, several rounds of axial coding will take place where emerging themes within and between participants will be labeled. Finally, the emergent themes will be integrated into an exhaustive representation that describes the overall experience of the families. Second, treatment attendance based upon registrations will be calculated. Finally, treatment adherence will be evaluated by calculating the extent to which the treatment protocol has been executed by the therapists based on ratings by independent raters.

## Data management

The data will be saved under administration numbers to which each family member is linked and stored on the protected ICT environment of the University of Amsterdam. The data analysis will be performed at the University of Amsterdam. Personal data will be handled conform the General Data Protection Regulation.

## Discussion

Ample support has been found for the notion that adolescent's mental health problems are rooted in, and exacerbated by, insecure attachment relations characterized by dysfunctional

emotional regulation within families. EFFT aims at developing secure attachment and functional emotional regulation. The current pilot study will be the first to examine feasibility and the potential effectiveness of EFFT. We adapted the original EFFT as described by Furrow et al. [5] and developed structured EFFT to reduce complexity inherent to family interventions.

Notable strengths of this study are the repeated measurements and the inclusion of a waiting period and a follow-up. The waiting period is important in examining spontaneous remission (reduction of complaints and insecure attachment without treatment), whereas the follow-up is important to get an impression of post-treatment maintenance of treatment gain. Repeated measurements are important to get more detailed insight into change over time and to enhance statistical power.

An important limitation of the study is the fact that its design is uncontrolled. This means that change during treatment cannot be attributed with 100% certainty to the intervention as spontaneous remission remains a possible explanation. To account for this, we included a waiting period. When change during treatment is larger than change during the waiting period it is likely it can be attributed to EFFT. Other limitations are the limited size of the convenience sample recruited by private practices, meaning that generalizability over various disorders should be interpreted with caution. Therefore, we plan a larger controlled study in the future to assess the effectiveness of EFFT as a transdiagnostic intervention for families.

## Supporting information

**S1 File. Roadmap structured EFFT.**
(DOCX)

**S1 Appendix. SPIRIT checklist.**
(DOCX)

**S1 Protocol. Effectiveness and feasibility of structured emotionally focused family therapy: A pilot study.**
(PDF)

## Acknowledgments

We would like to thank the participating family members in advance.

## Author Contributions

**Conceptualization:** Henk Jan Conradi, Lenny Rodenburg, Pieter Dingemanse, Trudy Mooren.

**Methodology:** Henk Jan Conradi, Daphne Meuwese, Trudy Mooren.

**Writing – original draft:** Henk Jan Conradi, Trudy Mooren.

**Writing – review & editing:** Henk Jan Conradi, Daphne Meuwese, Lenny Rodenburg, Pieter Dingemanse, Trudy Mooren.

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
