## [Decision Letter · Decision Letter 0]

25 Jan 2023

PONE-D-22-28187Effectiveness and feasibility of structured emotionally focused family therapy for parents and adolescents: Protocol of a within-subjects pilot studyPLOS ONE

Dear Dr. Conradi,

Thank you for submitting your manuscript to PLOS ONE. After careful consideration, we feel that it has merit but does not fully meet PLOS ONE’s publication criteria as it currently stands. Therefore, we invite you to submit a revised version of the manuscript that addresses the points raised during the review process.

Specifically, much more detail is requested regarding the protocol and outcome measures. 

We look forward to receiving your revised manuscript.

Kind regards,

Kymberly D. Young, Ph.D.

Academic Editor

PLOS ONE

and https://journals.plos.org/plosone/s/file?id=ba62/PLOSOne_formatting_sample_title_authors_affiliations.pdf.

“PD is member of the board of EFT Netherlands. LR provides EFFT. The other authors declare that they have no

competing interests.”

3. Please ensure that you include a title page within your main document. You should list all authors and all affiliations as per our author instructions and clearly indicate the corresponding author.

4. Please amend your authorship list in your manuscript file to include author list.

5. We note that the original protocol file you uploaded contains a confidentiality notice indicating that the protocol may not be shared publicly or be published. Please note, however, that the PLOS Editorial Policy requires that the original protocol be published alongside your manuscript in the event of acceptance. Please note that should your paper be accepted, all content including the protocol will be published under the Creative Commons Attribution (CC BY) 4.0 license, which means that it will be freely available online, and any third party is permitted to access, download, copy, distribute, and use these materials in any way, even commercially, with proper attribution.

Therefore, we ask that you please seek permission from the study sponsor or body imposing the restriction on sharing this document to publish this protocol under CC BY 4.0 if your work is accepted. We kindly ask that you upload a formal statement signed by an institutional representative clarifying whether you will be able to comply with this policy. Additionally, please upload a clean copy of the protocol with the confidentiality notice (and any copyrighted institutional logos or signatures) removed.

6. We note that the original protocol that you have uploaded as a Supporting Information file contains an institutional logo. As this logo is likely copyrighted, we ask that you please remove it from this file and upload an updated version upon resubmission.

7. Your ethics statement should only appear in the Methods section of your manuscript. If your ethics statement is written in any section besides the Methods, please delete it from any other section.

Additional Editor Comments:

The reviewer asks for much more detail about the protocol and measures.

Additionally, please address the following in your response:

Why was a RCT deemed not feasible for the pilot study? A between group comparison to the unmodified therapy would be a much stronger design. Do these modifications add any benefit? I believe this is a key question that is unanswered by the current design.

There are multiple follow-up measures, but it is unclear what the primary outcome is. Please clarify and justify.

Reviewers' comments:

Reviewer's Responses to Questions

**Comments to the Author**

1. Does the manuscript provide a valid rationale for the proposed study, with clearly identified and justified research questions?

Reviewer #1: Partly

2. Is the protocol technically sound and planned in a manner that will lead to a meaningful outcome and allow testing the stated hypotheses?

Reviewer #1: Partly

3. Is the methodology feasible and described in sufficient detail to allow the work to be replicable?

Reviewer #1: No

4. Have the authors described where all data underlying the findings will be made available when the study is complete?

Reviewer #1: No

5. Is the manuscript presented in an intelligible fashion and written in standard English?

Reviewer #1: Yes

6. Review Comments to the Author

You may also provide optional suggestions and comments to authors that they might find helpful in planning their study.

Reviewer #1: Reviewers comments

The protocol article presents a promising study within a growing field. The authors present a good rationale for their study and describe the study and the method fairly well. However, I have some comments. The reasons for my answers to the questions above are stated here.

1. Please be consistent throughout the manuscript regarding how many time points you are going to collect data at. It looks as if you first argue for three, then five, and in Table 1, I find six.

2. In the second sentence, you mention anger as a diagnostic category. Anger is definitely an important emotion. However, it is not a diagnosis in the DSM-5 or ICD-10.

3. I would like a reference the first time EFFT is introduced. Further, since there is another EFFT that seems to be very much like this one (both are presented in the book you refer to, actually in the following chapter, and both have almost identical names), I would like a comment so that there is no confusion as to which of the EFFTs you are planning to investigate, and thus provide a reason for why you are not referring to any of the studies of the other EFFT, which has several studies to refer to.

4. Please state more clearly that the structured EFFT you are to study differs from EFFT as described by the inventor and is thus something you have constructed.

5. Please provide a reason why this type of EFFT can be applied to parents and their children aged six and older. Why not younger?

6. On page 4, you mention the newly developed protocol of structured EFFT. Please mention there that it is provided further down in the manuscript.

7. Hypotheses: The first one (no, or less change) is two different hypotheses.

8. Method section. The last sentence in the first paragraph of the passage "design": "we will get a clear impression." Please restate. You do not know for sure that it will provide a clear impression.

9. Same passage: you state: "treatment protocol by the therapists will be based on structured observation checklists." I cannot find those structured checklists.

10. Participant section: it is not clear from the first sentence who will be included. I find it possible to read that you plan two groups. one with adolescents (12–18) and another one with families coping with mild problems.

11. Participant section. You write, "(c) families of which the parents or children are diagnosed with severe DSM disorders (substance abuse or psychosis) or are in therapy." I don't think it's fair to exclude parents and children with severe DSM diagnoses, as well as anyone in therapy for any reason. Please clarify.

12. Procedure and assessments section: You write, "to ensure 100% response rate to the other assessments, respondents will gather in the therapist’s room 20 minutes prior to the specific session to complete the questionnaires." This will not ensure a 100% response, but it will make it more probable.

13. Instruments section: You state in your description of the RDS that one unpublished study found a Chronbach's alpha of 72. That is an acceptable value, but not a good one. However, since the study is not yet published, it is not clear if that is related to the reworded version used in this pilot study. I believe it is not so. If it is, please specify. If not, argue for why the mentioned alpha value is relevant.

14. Instruments section: When describing the ARE you mention alpha.90 for men and women. In my reading of the referred paper, the alphas does not seem to be the same as once reported here. Further, I do not know if it is the abbreviated version that you are going to use that is presented there. and possibly not the reworded one you are going to use. Please explain.

15. Instruments section: When presenting the SEPTI-TS no alpha values are presented, only that they are acceptable. They are presented in the referred study.

16. Instruments section: When presenting the ECR-RS you refer to an unpublished manuscript for alpha values. Do they not even have pre-prints? I suggest you at least write that you will provide alpha values when you have the data. The same is true for CSI-4. I would like a reason for why you chose to use measures that are not solely tested.

17. At page 12, I am presented with two additional research questions (in the instruments section of the manuscript). I would like them to be presented earlier. You also mention "the developers of the EFFT treatment protocol." Please specify which of the authors this refers to. You also state that the researchers will conduct the interviews. Can you tell me how many and possibly who they are?

18. Statistical Analysis Section: It is a good statistical model you are suggesting to use. However, it is a very complex model for a low-N pilot study. One suggestion would be to remove the dyadic level and use it as a predictor instead. My suggestion is based on the fact that the specified model with three levels uses two levels on time. Further, I agree that there are no agreed upon power analyses concerning multilevel models with three levels, but to argue that you therefore estimate power by using a repeated-measures ANOVA is not convincing. One option is to simulate or use STATA to do the power analysis. Why not plan to run a repeated measures ANOVA on this pilot study and then only test the three-level multilevel analysis in this pilot study to prepare for a later large scale study? Further, I find the numbers you are using in the power analysis to be very optimistic. It is better to plan a study from a less optimistic perspective. You refer to two studies with large effect sizes. Reference no. 35 is a large study of ABFT (n = 341) that uses other measures than those used in this study. The high effect size stems from the self-reported suicidal ideation measure. Please provide an argument for why it is relevant. Reference no. 36 is a pilot study with only 11 participants in the treatment group (ABFT), and the effect size of 1.08 was only for one of the measures (the HAM-D). Again, please provide an argument for its relevance.

19. I did not get access to the SPIRIT diagram.

20. The authors state that the data will be made publicly available, but not where.

7. PLOS authors have the option to publish the peer review history of their article (what does this mean?). If published, this will include your full peer review and any attached files.

Reviewer #1: No

---

## [Author Response · Author response to Decision Letter 0]

10 Mar 2023

Appendix: Our Response to the Reviewers

Please note that the line numbers in the responses below correspond with the line numbers in the manuscript when the function “ track changes” is viewable. 

Comments of the Editor

We sincerely thank the editor for the statement that the manuscript has merit. We hope that with the changes that we have made to it, the manuscript now fully meets PLOS ONE’s publication criteria. 

Comment 1: Please ensure that your manuscript meets PLOS ONE's style requirements, including those for file naming. The PLOS ONE style templates can be found at

Response: Thank you for these PDF’s, hopefully we have met the requirements now.

Comment 2: Thank you for stating the following in the Competing Interests section:

“PD is member of the board of EFT Netherlands. LR provides EFFT. The other authors declare that they have no competing interests.”

Response: We have included the revised Competing Interests statement in the cover letter. 

Comment 3: Please ensure that you include a title page within your main document. You should list all authors and all affiliations as per our author instructions and clearly indicate the corresponding author.

Response: We have included the title page in the main document. 

Revised text: (lines 1 – 22).

Comment 4: Please amend your authorship list in your manuscript file to include author list.

Response: It was not entirely clear what was meant with this comment, but we have included an author list where the author contributions are described. 

Revised text: (line 443): Author list: HJC, DM, LR, PD, and TM.

Comment 5: We note that the original protocol file you uploaded contains a confidentiality notice indicating that the protocol may not be shared publicly or be published. Please note, however, that the PLOS Editorial Policy requires that the original protocol be published alongside your manuscript in the event of acceptance. Please note that should your paper be accepted, all content including the protocol will be published under the Creative Commons Attribution (CC BY) 4.0 license, which means that it will be freely available online, and any third party is permitted to access, download, copy, distribute, and use these materials in any way, even commercially, with proper attribution.

Therefore, we ask that you please seek permission from the study sponsor or body imposing the restriction on sharing this document to publish this protocol under CC BY 4.0 if your work is accepted. We kindly ask that you upload a formal statement signed by an institutional representative clarifying whether you will be able to comply with this policy. Additionally, please upload a clean copy of the protocol with the confidentiality notice (and any copyrighted institutional logos or signatures) removed.

Response: We do not know to which document this comment refers, we give permission to share the present study protocol paper. We have also included a treatment roadmap (S1. Roadmap structured EFFT).

Comment 6: We note that the original protocol that you have uploaded as a Supporting Information file contains an institutional logo. As this logo is likely copyrighted, we ask that you please remove it from this file and upload an updated version upon resubmission.

Response: We have removed the logo from the Spirit Checklist document. 

Comment 7: Your ethics statement should only appear in the Methods section of your manuscript. If your ethics statement is written in any section besides the Methods, please delete it from any other section.

Response: We have deleted line numbers 446-448 stating Ethics approval and consent to participate

Ethical approval has been obtained from the Ethics Review Board of the University of Amsterdam (2022-CP-15102).

Comment 8: Why was a RCT deemed not feasible for the pilot study? A between group comparison to the unmodified therapy would be a much stronger design. Do these modifications add any benefit? I believe this is a key question that is unanswered by the current design.

Response: We thank the editor for raising this issue, however, we did not intend to modify the original treatment in order to improve it. Our main focus is to study effectiveness and feasibility of EFFT. For this we added more structure to the original EFFT to make the treatment more suitable for research, i.e., more comparable between therapists. An RCT with the unmodified therapy as a control group would answer a different research question. We agree that an RCT with a waiting list condition would provide us with a stronger design that the current within-subjects design with waiting period we opted for. However, for this we first need to know whether structured EFFT is associated with improvement and with what effect size in order to design an adequate RCT.

Comment 9: There are multiple follow-up measures, but it is unclear what the primary outcome is. Please clarify and justify.

Response: We thank the editor for bringing this to our attention. Central to EFFT is the transdiagnostic idea that the adolescents problems are originating in and/or exacerbated by the negative interactions within the family (relational reframe). Therefore, we chose escalation into negative interactions between adolescent, mother and father as measured with the RDS as our primary outcome measure. The other measures are, important, secondary measures. We have clarified this in the text.

Revised text introduction: (lines 121 – 130): We chose escalation in negative interactions between child, mother and father as the primary outcome measure as the transdiagnostic idea that the child’s problems are originating in and/or exacerbated by negative interactions within the family is central to EFFT. Other equally important outcome measures are accessibility and responsiveness of the parents to their child’s needs; attachment between the parents and the child, and child psychological complaints. Additional outcome measures are: disciplining of the child by the parents; attachment between the parents as partners, and parental relationship satisfaction. Concerning all outcomes, we hypothesize:

Revised text method: (lines 258 – 268): Evaluation of effectiveness of EFFT in this pilot study focuses on the main objectives of the therapy: (1) de-escalation of negative interaction patterns; (2) enhancement of availability and responsiveness of the parents to the adolescent and between the parents as partners; (3) development of secure attachment between the adolescent and the parents and between the parents as partners; and (4) recovery from the adolescent’s psychological complaints that initiated seeking help. Additional outcome measures are: (5) enhancement of relationship satisfaction between the parents as partners; and (6) disciplining of the child by the parents. We opted for brief questionnaires measuring these concepts as several questionnaires will be completed twice during one assessment by the adolescent and/or parents (see below).

Comments Reviewer #1

We sincerely thank the reviewer for considering our study as promising and having a good rationale. 

Comment 1: Please be consistent throughout the manuscript regarding how many time points you are going to collect data at. It looks as if you first argue for three, then five, and in Table 1, I find six.

Response: We have 5 time points for the quantitative data with the last time point before the booster session. The qualitative data will be collected after the booster session, i.e., t6, which is why there are six timepoints mentioned in table 6. We clarified this in the text.

Revised text method: (lines 235 – 239): Therefore, it is important to enhance statistical power by applying five repeated assessments (t1 to t5) for the quantitative data: at pre-waiting period, pre-treatment, prior to phase 3, post-treatment, and prior to the booster session. Online and paper and pencil questionnaires will be administered. The qualitative data will be collected after the booster session (t6)

Comment 2: In the second sentence, you mention anger as a diagnostic category. Anger is definitely an important emotion. However, it is not a diagnosis in the DSM-5 or ICD-10.

Response: You are right, we have rephrased this sentence. 

Revised text: (line 52): These mental health difficulties may manifest themselves in many different diagnostic categories, such as depression, anxiety, withdrawal, addictions, and conduct-related problems.

Comment 3: I would like a reference the first time EFFT is introduced. Further, since there is another EFFT that seems to be very much like this one (both are presented in the book you refer to, actually in the following chapter, and both have almost identical names), I would like a comment so that there is no confusion as to which of the EFFTs you are planning to investigate, and thus provide a reason for why you are not referring to any of the studies of the other EFFT, which has several studies to refer to.

Response: We now include a reference the first time EFFT is introduced. Further, we are of course, willing to avoid confusion, but please help us by clarifying which ‘other’ EFFT you point at when you write: ‘Further, since there is another EFFT that seems to be very much like this one (both are presented in the book you refer to, actually in the following chapter, and both have almost identical names) …’

Revised text: (lines 59 – 61): Emotionally focused family therapy (EFFT) aims at the development of secure attachment between parents and their children to reduce children’s vulnerability to mental health problems.5

Comment 4: Please state more clearly that the structured EFFT you are to study differs from EFFT as described by the inventor and is thus something you have constructed.

Response: We have added a sentence describing that we have adapted the EFFT program to make it more suitable for research purposes. 

Revised text xxx: (lines 167 – 169): The protocol-driven intervention is an adapted version of the official EFFT program as described by Furrow et al.5 We adapted the EFFT program by applying more structure in order to make it more suitable for research purposes (see S1. Roadmap structured EFFT).

Comment 5: Please provide a reason why this type of EFFT can be applied to parents and their children aged six and older. Why not younger?

Response: We adhere to the EFFT guidelines. One can imagine that for EFFT enactments to work in the way they are described in the current guidelines, children need to be able to express their attachment needs verbally, which is imaginably difficult under the age of six.

Revised text xxx: (lines 110 – 112): This makes EFFT a promising systemic transdiagnostic intervention. EFFT can be applied with parents and their children aged six years and older, for children younger than six it would be difficult to express their attachment needs verbally.

Comment 6: On page 4, you mention the newly developed protocol of structured EFFT. Please mention there that it is provided further down in the manuscript.

Response: The protocol is provided by: (a) the summary in the text (‘Outline of the treatment protocol’), (b) the description in extenso in the original publication by Furrow et al. (2020) we refer at, and (c) the schema the therapists use (S1. Roadmap of structured EFFT) we now added.

Revised text Method: (lines 168 -169): We adapted the EFFT program by applying more structure in order to make it more suitable for research purposes (see S1. Roadmap structured EFFT).

Comment 7: Hypotheses: The first one (no, or less change) is two different hypotheses.

Response: We have changed the phrasing of the hypotheses. 

Revised text method: (line 131): (1) less change during the waiting period compared with the treatment phase;

Comment 8: Method section. The last sentence in the first paragraph of the passage "design": "we will get a clear impression." Please restate. You do not know for sure that it will provide a clear impression.

Response: We agree and have rephrased the sentence accordingly. 

Revised text xxx: (lines 141 – 143): By comparing change during the waiting period with change during the treatment phase, we probably get an impression of spontaneous remission vs. treatment-related change. 

Comment 9: Same passage: you state: "treatment protocol by the therapists will be based on structured observation checklists." I cannot find those structured checklists.

Response: The checklist is based on the roadmap for structured EFFT (S1.).

Revised text method: (lines 147 – 148): Finally, adherence to the treatment protocol by the therapists is based on the roadmap for structured EFFT (S1).

Comment 10: Participant section: it is not clear from the first sentence who will be included. I find it possible to read that you plan two groups. one with adolescents (12–18) and another one with families coping with mild problems.

Response: We have rephrased the sentence. 

Revised text method: (lines 150 – 152): For this pilot study we will include families with adolescents aged 12−18 years (the so called ‘identified’ patient) who are coping with mild problems.

Comment 11: Participant section. You write, "(c) families of which the parents or children are diagnosed with severe DSM disorders (substance abuse or psychosis) or are in therapy." I don't think it's fair to exclude parents and children with severe DSM diagnoses, as well as anyone in therapy for any reason. Please clarify.

Response: This study is a pilot study, which is why a certain homogeneity of the sample is important. Treatment is not dependent on research participation. Thus, we will not be excluding these participants from treatment, they will merely not be recruited for research. 

Revised text method: (lines 152 – 156): The latter means that we will exclude (a) blended families because of the more complex loyalties that exist between children and stepparents, (b) families of which individual members cope with serious trauma such as sexual and physical abuse and severe neglect, and (c) families of which the parents or children are diagnosed with severe DSM disorders (substance abuse or psychosis) or are in therapy to maintain sufficient research sample homogeneity for this pilot study.

Comment 12: Procedure and assessments section: You write, "to ensure 100% response rate to the other assessments, respondents will gather in the therapist’s room 20 minutes prior to the specific session to complete the questionnaires." This will not ensure a 100% response, but it will make it more probable.

Response: We agree and have rephrased this sentence. 

Revised text method: (lines 252 – 254): Second, to make a 100% response rate to the other assessments probable respondents will gather in the therapist’s room 20 minutes prior to the specific session to complete the questionnaires.

Comment 13: Instruments section: You state in your description of the RDS that one unpublished study found a Chronbach's alpha of 72. That is an acceptable value, but not a good one. However, since the study is not yet published, it is not clear if that is related to the reworded version used in this pilot study. I believe it is not so. If it is, please specify. If not, argue for why the mentioned alpha value is relevant.

Response: The alpha is related to the original version, the reworded version was not examined. The mentioned alpha is relevant as it assesses the same construct as the original version from a different perspective (i.e. the child instead of the parent). 

Revised text xxx: (lines 277 – 278): In an earlier study (under review) we found for the original version a Cronbach’s alpha of 0.72 for men and 0.74 for women.

Comment 14: Instruments section: When describing the ARE you mention alpha.90 for men and women. In my reading of the referred paper, the alphas does not seem to be the same as once reported here. Further, I do not know if it is the abbreviated version that you are going to use that is presented there. and possibly not the reworded one you are going to use. Please explain.

Response: You are right that the reported alpha’s in the referred paper (ranging between .94 and .96) are not the same as the one (.90) reported in the current manuscript. The .90 reported here is obtained from a recalculation of the data reported in the referred paper. The recalculation concerned the brief original version of the ARE, which will be used for the parents as partners relationship. It does not concern the reworded version which will be used for the parent child relationship. 

Revised text xxx: (lines 289 – 291): Cronbach’s alpha for the six-item scale is .90 for men and women, which is obtained from a recalculation of data earlier published.23 Of note, this concerns the original, not the reworded, version.

Comment 15: Instruments section: When presenting the SEPTI-TS no alpha values are presented, only that they are acceptable. They are presented in the referred study.

Response: You are right, we now present the alpha.

Revised text xxx: (line 299): Cronbach’s alpha is .81.24

Comment 16: Instruments section: When presenting the ECR-RS you refer to an unpublished manuscript for alpha values. Do they not even have pre-prints? I suggest you at least write that you will provide alpha values when you have the data. The same is true for CSI-4. I would like a reason for why you chose to use measures that are not solely tested.

Response: Actually, both measures are psychometrically tested and alphas for both are sound. We now refer to the original validation studies (Fraley et al., 2011 for the ECR-RS and Funk & Rogge, 2007 for the CSI) and report the alpha’s provided in these papers.

Revised text xxx: (lines 304 – 305 and 320): Cronbach’s alphas for Anxiety are 0.90 for men and 0.88 for women, and for Avoidance 0.90 for men and 0.92 respectively.25 

And concerning the CSI: Cronbach’s alpha is 0.94.26

Comment 17: At page 12, I am presented with two additional research questions (in the instruments section of the manuscript). I would like them to be presented earlier. You also mention "the developers of the EFFT treatment protocol." Please specify which of the authors this refers to. You also state that the researchers will conduct the interviews. Can you tell me how many and possibly who they are?

Response: The two feasibility question you refer at ((1) How do family members evaluate (specific phases of) EFFT?, and (2) What can we learn from their experiences?) are now introduced earlier in the introduction.

Revised text introduction: (lines 115 – 117): Evaluation of the feasibility of EFFT will be based on semi-structured interviews with family members who participated to obtain their evaluation of (specific phases of) EFFT and to identify possible points of improvement of EFFT. 

Response: The developers of the EFFT treatment protocol are now listed in the text: 

Revised text method: (lines 335 – 336): This list is the result of discussion within the research group, and the developers of the EFFT treatment protocol (LR, DM and HJC).

Comment 18: Statistical Analysis Section: It is a good statistical model you are suggesting to use. However, it is a very complex model for a low-N pilot study. One suggestion would be to remove the dyadic level and use it as a predictor instead. My suggestion is based on the fact that the specified model with three levels uses two levels on time. Further, I agree that there are no agreed upon power analyses concerning multilevel models with three levels, but to argue that you therefore estimate power by using a repeated-measures ANOVA is not convincing. One option is to simulate or use STATA to do the power analysis. Why not plan to run a repeated measures ANOVA on this pilot study and then only test the three-level multilevel analysis in this pilot study to prepare for a later large scale study? Further, I find the numbers you are using in the power analysis to be very optimistic. It is better to plan a study from a less optimistic perspective. You refer to two studies with large effect sizes. Reference no. 35 is a large study of ABFT (n = 341) that uses other measures than those used in this study. The high effect size stems from the self-reported suicidal ideation measure. Please provide an argument for why it is relevant. Reference no. 36 is a pilot study with only 11 participants in the treatment group (ABFT), and the effect size of 1.08 was only for one of the measures (the HAM-D). Again, please provide an argument for its relevance.

Response: We agree with the reviewer that the proposed multi-level model is complex. Although it reflects the complexity and dependencies within families and the multiple dyadic relationships therein, i.e., mother-adolescent, father-adolescent and mother-father as partners, it makes power estimation unsure. In order to make things more simple and transparent we follow your suggestion and will test our main hypothesis, i.e., the expectation of improvement during treatment on the primary outcome, with a repeated measurements ANOVA. This means that the power analysis based on a repeated measurements ANOVA we propose will correspond with our main analysis and will give a clear estimation of the expected power. 

Concerning the power analysis the reviewer argued that the numbers we used were too optimistic. Based on the mentioned two studies on ABFT (akin to EFFT) which reported effect sizes of .97 and 1.08 we assumed an effect size of .8 to be feasible. We agree this is a large effect size, however, earlier we achieved with a Couple Relationship Education course based on EFT-principles an average effect size on all outcome measures (relationship satisfaction, accessibility and responsiveness measured with the ARE, daily coordination between partners, relationship maintenance behavior, and forgiveness) of .6 in a self-referred sample (Conradi et al. 2017). Because EFFT is a therapy, not a course, we think it will not be too optimistic to assume an effect size of .7. However, we recalculated power with the assumption of this somewhat more conservative effect size of 0.7 instead of 0.8 and found that including 20 families will provide us with sufficient power. This is a conservative power-estimation based on the number of families as if we only had one person in each family reporting on the specific outcome measures. However, the adolescent will report, for example, twice per measurement point on the level of negative interactions (RDS), regarding the relation with the father and regarding the relation with the mother, whereas the mother will complete the RDS concerning the interactions with the adolescent and the father, and the father concerning the interactions with the adolescent and the mother. This means that the main effect of time concerning, for example, the RDS will be based on at maximum 480 observations (3 persons*2 measurements*4 measurement points*20 families). The more complex multilevel model uses the fact that we measure multiple individuals within families, while accounting for the dependency between observations between them.

Finally, we will follow the suggestion of the reviewer and after testing our main hypothesis with a repeated measurements ANOVA we will conduct the three-level multilevel analysis on the data of this pilot study. With such a model one can not only adjust for dependencies between the repeated measurements as is the case with repeated measurements ANOVAs, but also for dependency of measurements between people within dyads. When results of the multilevel model converge with the repeated measures ANOVAS, trust in the robustness of findings will be enhanced.

Revised text xxx: (lines 346 – 378):

As scores are not independent within respondents (repeated measurements) and between respondents (three dyads: mother and the adolescent, father and the adolescent, and the parents as partners) multi-level analyses are preferred. However, concerning multilevel models with three levels no agreed upon power analyses are available. Therefore, we estimated study power based on a repeated measurements ANOVA with GPower 3.1.9.7.30 We anticipate an effect size of d = 0.7 which is based on effect sizes obtained by Attachment-Based Family Therapy, which is akin to EFFT (i.e., d = .9731 and d = 1.0832), and the average effect size we obtained with a Couple relationship Education course based on EFT (i.e., d = 0.623). We computed the required sample size to detect this effect size of d = 0.7 with a power of .8 an alpha of .05 and a correction for sphericity of e=.5. It was calculated we would need n = 20 families. This is a conservative power-estimation based on the number of families as if we only had one person in each family reporting on the outcome measures. However, as several of our outcomes are dyadic by nature (e.g., outcomes regarding interaction patterns between mother and the adolescent, father and the adolescent, and father and mother as partners) the actual power will be higher as the number of individuals will be twice as high for these outcomes.

Following the power analysis we first plan to run a repeated measurements ANOVA to test our main hypothesis concerning change during treatment plus follow-up on the primary outcome. Subsequently, Linear Mixed Models (LMM) in SPSS will be applied with repeated measurements (level 1) nested within respondents (level 2) and, depending on the specific outcome, respondents nested within the dyad of interest (level 3).33 In this way we will be able to assess change over time. An advantage of LMM is the possibility to use cases with partial missing data. To adjust for interdependence of the repeated measurements, the AR1 covariance structure will be applied, and to adjust for interdependence of respondents, we will include an intercept at level 3. When results of the multilevel model converge with the repeated measures ANOVA, trust in the robustness of findings will be enhanced.

We will examine change during the waiting period (comparison of pre-treatment versus pre-waiting period assessments), change during treatment (post-treatment versus pre-treatment assessments), change during follow-up (booster session versus post-treatment assessments), and change during the whole study period (follow-up versus pre-waiting period assessments). We will compute effect sizes (Cohen’s ds) using, per comparison, the estimated marginal means, and the pooled standard deviations of the corresponding raw means.34 Effect sizes will be interpreted as small when Cohen’s d is 0.35 or below, moderate when 0.36−0.65, and large when 0.66 or higher.35 To evaluate missingness at random, we will apply pattern-mixture models for the dependent variables.36

Comment 19: I did not get access to the SPIRIT diagram.

Response: We added the SPIRIT diagram again. Hopefully, now it will work properly.

Comment 20: The authors state that the data will be made publicly available, but not where.

Response: This is a protocol paper, consequently it does not report on any data. We will state where the data will be made publicly available in the paper that presents data.

---

## [Decision Letter · Decision Letter 1]

21 Mar 2023

PONE-D-22-28187R1Effectiveness and feasibility of structured emotionally focused family therapy for parents and adolescents: Protocol of a within-subjects pilot studyPLOS ONE

Dear Dr. Conradi,

Thank you for submitting your manuscript to PLOS ONE. After careful consideration, we feel that it has merit but does not fully meet PLOS ONE’s publication criteria as it currently stands. Therefore, we invite you to submit a revised version of the manuscript that addresses the points raised during the review process.

 Please respond in detail to each of the reviewer's remaining comments. 

We look forward to receiving your revised manuscript.

Kind regards,

Kymberly D. Young, Ph.D.

Academic Editor

PLOS ONE

Journal Requirements:

Reviewers' comments:

Reviewer's Responses to Questions

**Comments to the Author**

1. Does the manuscript provide a valid rationale for the proposed study, with clearly identified and justified research questions?

Reviewer #1: Yes

2. Is the protocol technically sound and planned in a manner that will lead to a meaningful outcome and allow testing the stated hypotheses?

Reviewer #1: Yes

3. Is the methodology feasible and described in sufficient detail to allow the work to be replicable?

Reviewer #1: Yes

4. Have the authors described where all data underlying the findings will be made available when the study is complete?

Reviewer #1: No

5. Is the manuscript presented in an intelligible fashion and written in standard English?

Reviewer #1: Yes

6. Review Comments to the Author

You may also provide optional suggestions and comments to authors that they might find helpful in planning their study.

Reviewer #1: I am pleased with how the authors addressed the majority of my initial comments. I apologize for not being explicit enough in a few of my initial comments and will clarify them so that the authors can comment and/or revise.

Comment 3

There are two interventions with the abbreviation EFFT. The Encyclopedia of Couple and Family Therapy presents the two in consecutive chapters. Emotionally focused family therapy by Furrow and Palmer and Emotion-focused family therapy by Sabey and Lafrance are the two distinct EFFTs. Many years of research support the latter. See e.g., https://efftinternational.org/research.

Since these two EFFTs are available today and the latter has been investigated for a number of years, I would propose that you include a section in the manuscript that comments on this so that the readers are informed and there is no misunderstanding as to which EFFT you intend to explore. If you do not include any of the research on that EFFT in your manuscript, you should also provide a reason why it is not relevant.

Comment 11

I appreciate your clarification. I want to confirm that I and future readers have correctly understood your meaning. Do you intend to exclude any children or parents who are undergoing therapy for whatever reason?

Comment 20 

Under the guidelines for me as a reviewer, PLOS ONE asks, "Have the authors described where all data underlying the findings will be made available when the study is complete?"Since you have not stated WHERE it will be made available, I have to answer "no" to that question. I suggest you state where it is supposed to be publicly stored.  One suggestion is osf.io, and another is data.mendeley.com, but you are free to choose any other repositories.

7. PLOS authors have the option to publish the peer review history of their article (what does this mean?). If published, this will include your full peer review and any attached files.

Reviewer #1: **Yes: **Rune Zahl-Olsen

---

## [Author Response · Author response to Decision Letter 1]

3 May 2023

PONE-D-22-28187R1

Effectiveness and feasibility of structured emotionally focused family therapy for parents and adolescents: Protocol of a within-subjects pilot study

PLOS ONE

Reviewers' comments:

Reviewer's Responses to Questions

Comments to the Author

6. Review Comments to the Author

You may also provide optional suggestions and comments to authors that they might find helpful in planning their study.

Reviewer #1: I am pleased with how the authors addressed the majority of my initial comments. I apologize for not being explicit enough in a few of my initial comments and will clarify them so that the authors can comment and/or revise.

Comment 3

There are two interventions with the abbreviation EFFT. The Encyclopedia of Couple and Family Therapy presents the two in consecutive chapters. Emotionally focused family therapy by Furrow and Palmer and Emotion-focused family therapy by Sabey and Lafrance are the two distinct EFFTs. Many years of research support the latter. See e.g., https://efftinternational.org/research.

Since these two EFFTs are available today and the latter has been investigated for a number of years, I would propose that you include a section in the manuscript that comments on this so that the readers are informed and there is no misunderstanding as to which EFFT you intend to explore. If you do not include any of the research on that EFFT in your manuscript, you should also provide a reason why it is not relevant.

Reply 3

You are right. We were aware of Emotion-focused family therapy by Lafrance et al. (2020), but were not sure you meant this particular EFFT. We made clear in the text it should not be confused with Emotionally-focused family therapy originating with Furrow and Palmer (2019). The latter has the attachment relationship as a more central focus. Although important, we did not include the Emotion-focused family therapy studies you referred at as these studies concern the brief group course based on Emotion-focused therapy and not the regular therapy and as such is not comparable to EFFT examined in this study. We now write (lines 93-96):

Of note, Emotionally-focused family therapy, the intervention examined in this study, should not be confused with Emotion-focused family therapy.18 Emotionally-focused family therapy has attachment as its central focus, whereas Emotion-focused family therapy works with the general emotion coaching principles as described by Greenberg.

Comment 11

I appreciate your clarification. I want to confirm that I and future readers have correctly understood your meaning. Do you intend to exclude any children or parents who are undergoing therapy for whatever reason?

Reply 11

Yes.

Comment 20 

Under the guidelines for me as a reviewer, PLOS ONE asks, "Have the authors described where all data underlying the findings will be made available when the study is complete?" Since you have not stated WHERE it will be made available, I have to answer "no" to that question. I suggest you state where it is supposed to be publicly stored. One suggestion is osf.io, and another is data.mendeley.com, but you are free to choose any other repositories.

Reply 20

We followed your suggestion (line 417).

---

## [Editor Report · Decision Letter 2]

6 Jun 2023

Effectiveness and feasibility of structured emotionally focused family therapy for parents and adolescents: Protocol of a within-subjects pilot study

PONE-D-22-28187R2

Dear Dr. Conradi,

We’re pleased to inform you that your manuscript has been judged scientifically suitable for publication and will be formally accepted for publication once it meets all outstanding technical requirements.

Kind regards,

Kymberly D. Young, Ph.D.

Academic Editor

PLOS ONE
---

## [Editor Report · Acceptance letter]

15 Jun 2023

PONE-D-22-28187R2 

Effectiveness and feasibility of structured emotionally focused family therapy for parents and adolescents:
Protocol of a within-subjects pilot study 

Dear Dr. Conradi:

I'm pleased to inform you that your manuscript has been deemed suitable for publication in PLOS ONE. Congratulations! Your manuscript is now with our production department. 

Kind regards, 

on behalf of

Dr. Kymberly D. Young 

Academic Editor

PLOS ONE